# Designed folding pathway of modular coiled-coil-based proteins

Jana Aupič[1,7], Žiga Strmšek [1,2,7], Fabio Lapenta [1,3], David Pahovnik[4], Tomaž Pisanski [5,6], Igor Drobnak[1], Ajasja Ljubetič [1] & Roman Jerala [1,3✉]

Natural proteins are characterised by a complex folding pathway defined uniquely for each fold. Designed coiled-coil protein origami (CCPO) cages are distinct from natural compact proteins, since their fold is prescribed by discrete long-range interactions between orthogonal pairwise-interacting coiled-coil (CC) modules within a single polypeptide chain. Here, we demonstrate that CCPO proteins fold in a stepwise sequential pathway. Molecular dynamics simulations and stopped-flow Förster resonance energy transfer (FRET) measurements reveal that CCPO folding is dominated by the effective intra-chain distance between CC modules in the primary sequence and subsequent folding intermediates, allowing identical CC modules to be employed for multiple cage edges and thus relaxing CCPO cage design requirements. The number of orthogonal modules required for constructing a CCPO tetrahedron can be reduced from six to as little as three different CC modules. The stepwise modular nature of the folding pathway offers insights into the folding of tandem repeat proteins and can be exploited for the design of modular protein structures based on a given set of orthogonal modules.

[1] Department of Synthetic Biology and Immunology, National Institute of Chemistry, Ljubljana, Slovenia. [2] Interdisciplinary Doctoral Programme in Biomedicine, University of Ljubljana, Ljubljana, Slovenia. [3] EN-FIST Centre of Excellence, Ljubljana, Slovenia. [4] Department of Polymer Chemistry and Technology, National Institute of Chemistry, Ljubljana, Slovenia. [5] FAMNIT, University of Primorska, Koper, Slovenia. [6] Institute of Mathematics, Physics and Mechanics, Ljubljana, Slovenia. [7] These authors contributed equally: Jana Aupič, Žiga Strmšek. ✉email: roman.jerala@ki.si

Proteins are the most versatile type of polymers that fold into diverse structural folds and underlay almost all biological functions. Protein design aspires to bring about the development of new protein scaffolds and functional proteins tailor-made for specific applications. Designing proteins from first principles requires detailed knowledge of complex and manifold interactions that dictate protein folding and self-assembly, as well as significant computational power[1–3]. Hitherto, computational protein design has been successfully applied to the design of proteins containing up to ~120 amino acid residues in a single polypeptide chain[4,5]. Modular protein design aims to simplify the construction of protein architectures by employing well-understood polypeptide modules, such as coiled-coils, as building blocks[6–11]. Coiled-coils (CC) are a frequent suprastructural element, characterized by a heptad repeat pattern customarily denoted as abcdefg[12]. The assembly of two or more peptide chains into a left-handed superhelix is mediated by hydrophobic and electrostatic interactions between amino acids at a, d and e, g positions, respectively (Fig. 1a). A relatively straightforward sequence-structure relationship has allowed for the successful design of CCs with specific oligomerization number, peptide chain orientation and interaction specificity[13]. It has been recently shown that by concatenating coiled-coil dimer forming peptides in a defined order into a single polypeptide chain modular polyhedral cage-shaped protein folds can be designed (Fig. 1b)[9,10]. This approach, termed coiled-coil protein origami (CCPO) as it employs a similar modular strategy as DNA nanotechnology[14,15], relies on the availability of an orthogonal set of CC dimers, i.e. sets of peptides where each peptide forms a dimer only with its cognate peptide partner. In contrast to DNA, where the design of orthogonal DNA duplexes is rather trivial due to the straightforward nucleotide pairing rules, currently available orthogonal CC sets are limited in size to a dozen or so validated orthogonal elements[16–20] (Fig. 1c, d, Supplementary Table 1), hindering the achievable complexity of CCPO cages. Their design is further complicated by the fact that for most CCPO cage architectures topological rules require the use of both parallel and antiparallel CCs. Hitherto, the design of CCPO folds was based on the assumption that the orthogonality of building modules is a condition sine qua non for the successful design of CCPO polyhedra, however this might not be the case.

Domain repeats are commonly observed in natural proteins; in fact tandem repeat proteins represent 20 % of the proteome in eukaryotes[21] and are implicated in signalling, cell-adhesion, complex assembly[22,23] as well as in several human disorders[24]. A particular problem for tandem repeat proteins is that interactions that stabilize the native fold may also stabilize misfolded states, arising from intra-chain domain-swapping[25]. Experiments on a model system, composed of two covalently linked immunoglobulin-like domains from the I-band of titin, revealed that domain-swapping can indeed result in long-lasting misfolded species[25,26]. Several experimental[27,28] as well as theoretical studies[29,30] have suggested that protein folding is often primarily determined by protein topology. Moreover, folding studies on small globular proteins showed that rate constants are inversely correlated with total contact order, which reflects the average distance between native contacts in the amino acid sequence[31]. What all of this means for folding of complex, multi-domain protein architectures such as CCPO cages is less clear. Since CC-forming peptides are positioned at varying intra-chain distances in CCPO cages, does their folding proceed in a modular stepwise manner? Furthermore, is the folding pathway governed by the intra-chain distance between interacting peptide building modules, or does their intrinsic thermodynamic stability and kinetic properties also play a role? Understanding the folding pathway of CCPO cages could not only elucidate the poorly studied folding

of modular protein structures, but would also be particularly useful for designing CCPO cages containing several instances of the same CC building blocks.

In this work, we investigate if the same CC building blocks could be used multiple times within the same single-chain tetrahedral design, without resulting in heterogeneous folds or misfolded structures (Fig. 1e). Molecular dynamics (MD) simulations based on the Gō force field[32] and stopped-flow folding kinetics measurements are used to establish the proximity between interacting CC peptides in the sequentially pre-organized structure as the major determinant of CCPO polyhedron folding. Based on the results we develop a mathematical model for predicting the folding probability of CCPO folds and apply it to the design of CCPO tetrahedra containing different numbers of CC building block repeats. We demonstrate that a single type of building module can indeed be used to assemble two edges of a polyhedron. One, two and as many as three building modules can be used twice in the same chain, decreasing the number of required building modules, thus expanding the complexity of modular polyhedra that could be constructed from a given set of building modules.

## Results

**Folding kinetics of CCPOs are governed by the intra-chain distance between interacting CC modules.** An accepted assumption in the design of modular single-chain protein assemblies is that orthogonal modules are needed to uniquely define the desired fold. On the other hand, employing multiple identical modules might be feasible if we could assure correct module pairing by accurately designing the assembly pathway of modular proteins. In order to determine the optimum positioning of repeating CC modules, we first performed MD folding simulations for several previously reported CCPO tetrahedron variants[10] using the all-atom Gō force field[32]. The Gō force field allows protein folding mechanisms to be examined at reduced computational cost by including an attractive term in the energy function to describe non-bonded interactions between atom pairs located in close proximity in the native structure. All investigated tetrahedral cage variants were composed of 12 concatenated peptide modules comprising 6 orthogonal CC pairs (4 parallel and 2 antiparallel), but were based on different topologies or circular permutations. Analysis of MD trajectories revealed that folding of each tetrahedral cage proceeded in multiple steps with each coiled-coil dimerization event occurring independently (Supplementary Movie 1–4). A CC pair was considered formed once 50% of native contacts were recapitulated. While multiple folding pathways were observed, the temporal order in which individual coiled-coils assembled was primarily determined by the spatial proximity of pairing peptide segments (Supplementary Figs. 1–3). To describe the latter we introduced the intra-chain distance metric, defined as the minimal number of peptide modules separating the termini of complementary peptide segments (Supplementary Fig. 1). In case of parallel CCs, the distance is calculated between matching termini (C-C or N-N), while for antiparallel CCs the distance is calculated between opposing segment ends (N-C). Once formed, a CC pair is counted as one segment (for detailed description see Supplementary Discussion). MD simulations indicated peptide pairs positioned at shorter intra-chain distances were more likely to fold first. In addition, we observed that coiled-coil forming peptides positioned at either terminal end exhibited a slightly higher folding rate than would be expected based on their intra-chain distance (Supplementary Fig. 2). This could be due to their higher degree of freedom in comparison to more centrally located peptide segments.

In contrast, previously reported global refolding kinetics suggested that CCPO cages fold according to a two-state model[10].

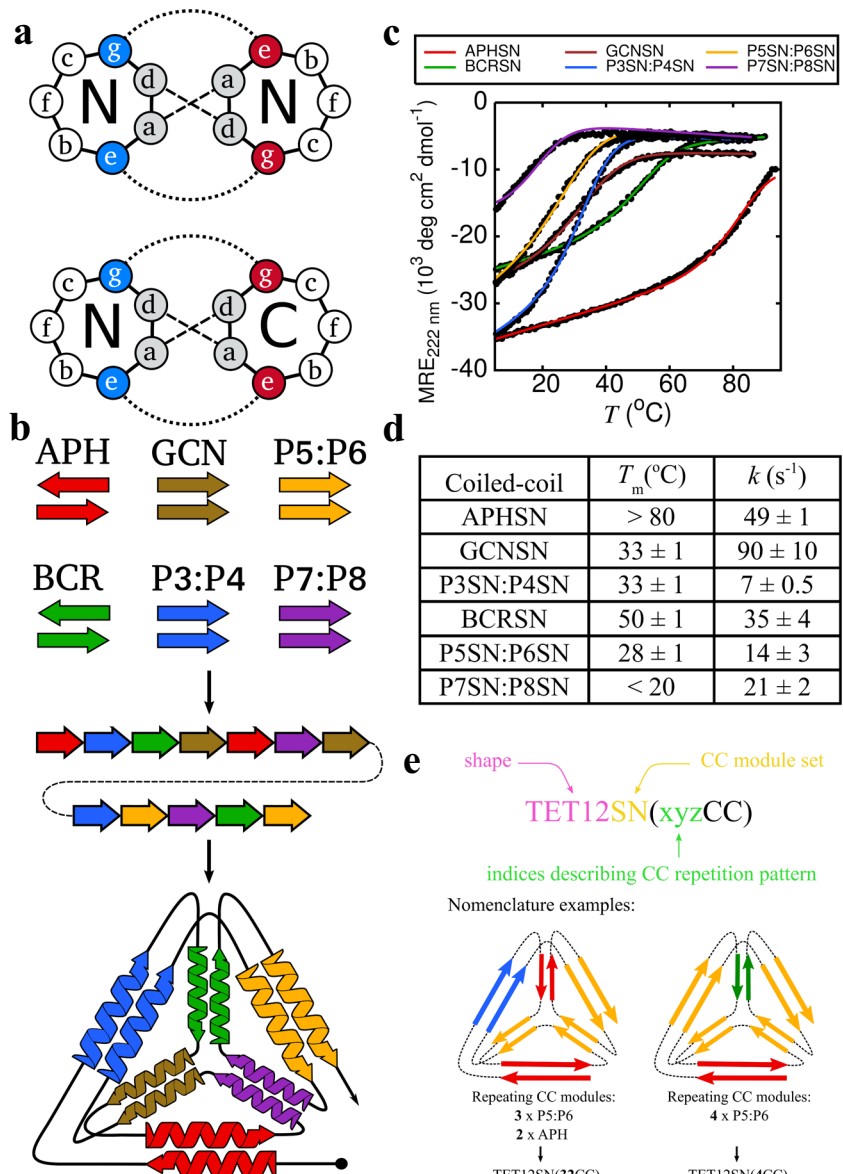

**Fig. 1 Coiled-coil protein origami (CCPO) design strategy. a** Helical wheel representation of interactions in parallel (above) and antiparallel coiled-coil (CC) dimers (below). **b** CCPO design strategy relies on covalently linking orthogonal CC dimer-forming peptides into a single polypeptide chain that folds into a polyhedral cage with CC dimers representing its edges. **c** Temperature unfolding curves for CC building modules comprising TET12SN. Melting temperatures were determined at 40 μM dimer concentration. **d** Melting temperatures ($T_m$) and rates of refolding ($k$) for CC building blocks. Initial concentrations of peptide dimers for stopped-flow experiments were 20 μM, resulting in 4 μM concentration post mixing. **e** Depiction of the naming convention applied to tetrahedral cage designs with repeated CC modules. CCPO cages are named according to the polyhedral shape they resemble (TET12), set of CC building blocks used for their construction (SN[10]) and the CC repetition pattern. Source data are provided as a Source Data file.

To experimentally confirm the prominent role of spatial proximity between interacting modules in determining the CCPO folding pathway observed in silico, we used multi-site Förster resonance energy transfer (FRET) to investigate the folding pathway of the previously designed tetrahedral cage TET12SN[10], composed of unique CC segments (Fig. 1b, see below). Initially, folding rates and melting temperatures of CC building modules in isolation were determined (Fig. 1c, d, Supplementary Figs. 4 and 5). Due to the different number of polar residues in the hydrophobic core and salt bridges between residues on e, g heptad positions (Supplementary Table 1), CC modules had different thermodynamic and kinetic stabilities (see Supplementary Discussion). In order to monitor folding and unfolding of individual edges within a tetrahedral cage, six TET12SN variants each with a pair of cysteine residues at a different investigated

edge were prepared by point mutagenesis (see Methods), isolated and chemically labelled with Sulfo-cy3 and Sulfo-cy5 via thiol-maleimide coupling.

First, equilibrium stability of TET12SN was determined by monitoring secondary structure as a function of guanidine hydrochloride (Gdn-HCl) concentration (Supplementary Fig. 6a) or temperature (see below). TET12SN exhibited a cooperative two-state unfolding with a denaturation midpoint at 2.5 M Gdn-HCl or 56 °C, respectively. On the other hand, chemical denaturation experiments with fluorescently labelled TET12SN variants revealed that the decrease in helical content is somewhat preceded by a loss of tertiary structure (Supplementary Fig. 6b). For most cage edges, the denaturation midpoint was observed at 2 M Gdn-HCl. The cage edge represented by the BCRSN module exhibited a transition midpoint at 1.5 M Gdn-HCl, while the

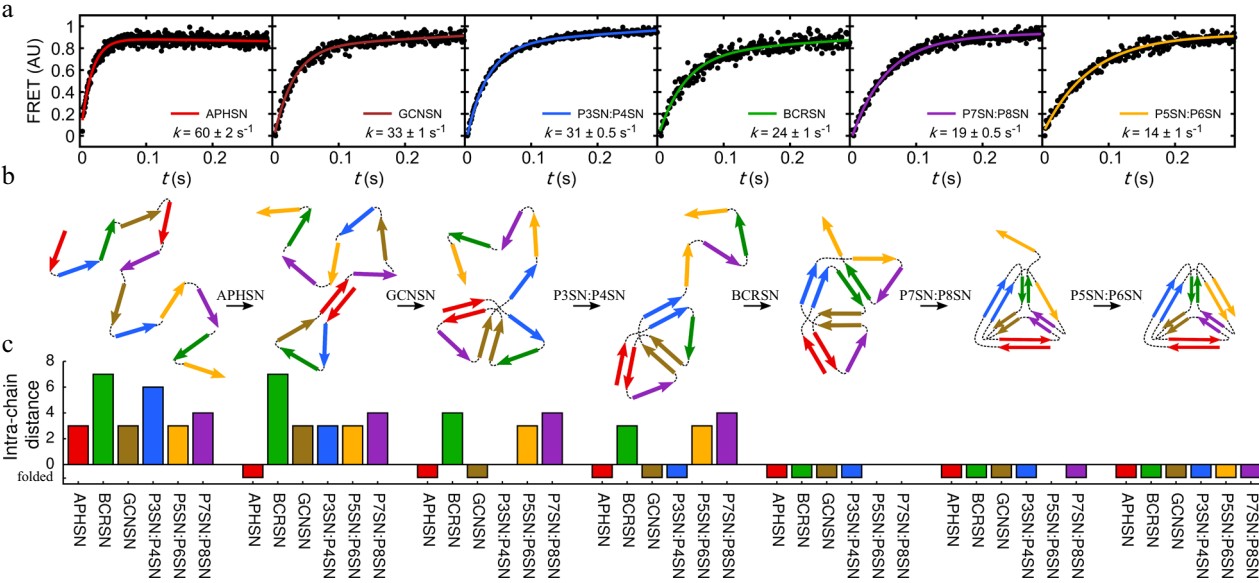

**Fig. 2 Stepwise modular folding mechanism of CCPO tetrahedron TET12SN. a** The order in which individual CC edges in the tetrahedral cage assemble was determined by comparing normalized time-resolved increase in acceptor fluorescence during refolding of TET12SN in 2 M Gdn-HCl observed for each of the different fluorescent dye placements. In each experiment, a pair of fluorescent dyes was conjugated to the appropriate pair of cysteine residues in the selected CC dimer allowing the folding of one edge of the tetrahedral cage to be tracked. Increase in Förster resonance energy transfer (FRET) was fit using a two-state kinetic model. FRET intensity is given in arbitrary units (AU). **b** Scheme of the proposed stepwise folding mechanism of TET12SN based on stopped-flow results shown in (**a**). **c** Changes in the effective intra-chain distance between peptide pairs during folding. In each step, one of the peptide pairs characterized by the shortest intra-chain distance formed. Source data are provided as a Source Data file.

unfolding transition of the APHSN module had a midpoint at ~3 M Gdn-HCl. Similarly, thermal denaturation scans revealed individual CC modules exhibit different melting temperatures ($T_m$). APHSN module exhibited two transitions with the main transition occurring at 45 °C, while other cage edges showed a single unfolding transition with $T_m$ values between 48 °C and 56 °C (Supplementary Fig. 7). Taken together, equilibrium experiments indicated that unfolding of TET12SN is a multistep process.

Next, folding kinetics were examined by unfolding fluorescently labelled TET12SN variants in 5 M Gdn-HCl followed by rapid dilution in the stopped-flow instrument, resulting in final Gdn-HCl concentration of 2 M. The observed increase in the FRET signal during refolding was fit to a two-state model. Individual CC modules of TET12SN exhibited significantly different kinetics (Fig. 2a). The dependence of refolding rates on the location of fluorescent probes suggested that TET12SN folds in stages (Fig. 2b), each corresponding to the folding of one of the six edges. The ranking order of folding rates ($k_F^{APHSN} = 60\ s^{-1} > k_F^{GCNSN} = 33\ s^{-1} > k_F^{P3SN:P4SN} = 31\ s^{-1} > k_F^{BCRSN} = 24\ s^{-1} > k_F^{P7SN:P8SN} = 19\ s^{-1} > k_F^{P5SN:P6SN} = 14\ s^{-1}$) did not match the rank of thermodynamic stabilities or refolding rates of individual CCs (Fig. 1c, d, Supplementary Figs. 4 and 5). On the other hand, the observed order of CC module assembly corresponded well to the changes in intra-chain distances accompanying the folding process (Fig. 2c). At each stage, the forming CC edge had or was among those with the shortest intra-chain distance between its peptide building blocks in the unfolded state or in the structural intermediate formed in the preceding folding step, indicating that the distance between CC building blocks, defined by chain topology, plays a dominant role in determining the folding pathway.

Since the experiments were carried out in the region of Gdn-HCl concentrations where the unfolding transition was observed,

any potential less stable intermediates might have been overlooked. Therefore, the experiments were carried out also at 1 M final Gdn-HCl concentration. The temporal order of CC module formation remained unchanged, however the folding constant of the APHSN module could not be reliably determined (Supplementary Figs. 8–10, see Supplementary Discussion).

**Design and modelling of CCPO cages comprising multiple copies of identical building modules.** The prominent role of intra-chain distance in determining the kinetics of segment assembly implied that we might be able to design CCPO cages containing CC module repetitions by guiding the interaction preference of repeating CC peptides through their positioning in the polypeptide sequence. Based on the results of folding experiments, a simplified deterministic model of folding (Supplementary Fig. 11) was devised to describe the stepwise transition of the CCPO cage from the unfolded (Supplementary Fig. 11a, I) to the folded state (Supplementary Fig. 11a, VIII). According to the model, at each folding step the peptide pair with the shortest intra-chain distance assembles into a CC dimer. After each folding event, the intra-chain distances are recalculated to account for the change in the effective distance between interacting peptide pairs in the intermediate structure due to CC pair formation. In case several peptide pairs are at an equal distance, the folding pathway is split and all possible paths are further examined (Supplementary Fig. 11a, II, III, IV). Peptide segments are treated as rigid bodies, prohibiting certain folding events due to steric constraints (Supplementary Fig. 12). The probability for a certain polypeptide sequence to fold correctly, $P_F$, was calculated as the ratio between the number of folding pathways ending in a tetrahedral cage and all possible folding pathways (Supplementary Fig. 11b). For any polypeptide sequence composed of unique coiled-coil building pairs the $P_F = 1$. However, multiple use of the same CC module in the polypeptide sequence could

lead to unproductive misfolded states (Supplementary Fig. 11a, VII) that cannot continue towards a tetrahedral structure and thus result in a decrease in the $P_F$.

**Characterisation of CCPOs containing multiple identical interacting modules.** The above described sequential assembly model was used to design CCPO tetrahedra where either one (TET12SN(2CC)) or two (TET12SN(22CC)) CC modules were used twice in the polypeptide sequence, decreasing the number of different CC modules from 6 to 5 or 4 (Fig. 3a–c). $P_F$ was calculated for all possible circular permutations and arrangements of peptide building blocks (Supplementary Tables 2–5) that could theoretically assemble into a tetrahedral cage[33]. In both design cases, several sequences were predicted to fold with $P_F = 1$. Among these, designs based on the circular permutation 1.10 were chosen for experimental validation, in order to facilitate comparison to the previously designed TET12SN (Fig. 3a)[10]. It is important to note here that the outcome of folding simulations depended solely on the sequential arrangement of modular building blocks in the polypeptide sequence while the underlying amino acid sequence of peptide building blocks had no bearing on the $P_F$. Building blocks for constructing the amino acid sequences were chosen to match those in TET12SN. CCs with higher stability were selected for the repetition, eliminating less stable modules (Fig. 1c). In addition, preference was given to heterodimeric pairs, since using multiple copies of homodimeric coiled-coils is more likely to lead to misfolding. Based on these considerations, TET12SN(2CC) contained two instances of P5SN: P6SN in the polypeptide sequence (Fig. 3b), while TET12SN (22CC) had two repeats of P5SN:P6SN and P3SN:P4SN (Fig. 3c).

Designed proteins were expressed in *E. coli* and isolated using Ni-NTA and size exclusion chromatography (SEC) (Supplementary Fig. 13). As TET12SN, both were highly helical (85–90%, Fig. 3d–f) and displayed cooperative unfolding (Fig. 3g–i). In comparison to TET12SN ($T_m = 56\,°C$), TET12SN(2CC) displayed a somewhat lower $T_m$ (52 °C), most likely due to the replacement of the GCNSN module with the less stable P5SN: P6SN pair (Fig. 1c, Supplementary Fig. 7). Similarly, substituting P7SN:P8SN with the more stable P3SN:P4SN module in TET12SN(22CC) led to an increase in $T_m$ (56 °C). Thermal unfolding was reversible and the proteins were able to efficiently refold upon cooling (Fig. 3d–f, Supplementary Fig. 14). We hypothesised that domain-swapping between repeated CCs would prevent all of the CCs to assemble in the context of a single chain and would therefore give rise to a population of partially unfolded states and subsequent formation of larger aggregates. Sample polydispersity was analysed with SEC coupled to multi-angle light scattering (MALS) measurements, where the observed degree of aggregation upon refolding was negligible and comparable to that of TET12SN[10] (Fig. 3j–l), indicating domain-swapping did not occur on a significant scale. Moreover, thermal denaturation scans were repeated after refolding and showed no change in $T_m$ (Supplementary Fig. 14).

Establishing that heterodimeric parallel building blocks may be used twice in the same polypeptide chain, we next sought to duplicate the antiparallel homodimer, resulting in a further reduction to only three unique CC pairs (TET12SN(222CC)) (Fig. 4a). In addition, we investigated whether a single CC dimer could be repeated three times within the polypeptide sequence (TET12SN(3CC)) (Fig. 4b). TET12SN(222CC) was based on the circular permutation 1.10 and was constructed from P3SN:P4SN, P5SN:P6SN and the antiparallel homodimer APHSN, each of which was used for two CCPO cage edges (Supplementary Tables 6 and 7). Conversely, in case of TET12SN(3CC), all sequences based on the circular permutation 1.10 had a $P_F$ value

below 1, therefore circular permutation 1.5, which had the highest $P_F$, was selected as the basis for the design (Supplementary Tables 8 and 9). To better assess the validity of the proposed folding model, the arrangement of building blocks with the lowest $P_F$ (TET12SN(3CC)-neg) was experimentally characterised as well (Fig. 4c, Supplementary Table 9). The peptide pair P5SN: P6SN was selected as the repeating module. While all proteins could be obtained in monomeric form, the yield for TET12SN (3CC) and TET12SN(3CC)-neg was markedly reduced (Supplementary Fig. 15). Nonetheless, all cage designs exhibited high helicity (Fig. 4d–f) and two transitions during thermal unfolding (Fig. 4g–i). This is most likely due to the greater number of repeating CC modules, increasing their contribution to the thermal unfolding profile, resulting in individual transitions becoming more resolved. In case of TET12SN(222CC) the first transition occurred at 44 °C, which corresponds well to the melting temperature of the main unfolding transition observed for the APHSN module in the context of the tetrahedral cage (Supplementary Fig. 7). After rapid refolding, the amount of higher oligomeric-order species observed for TET12SN(222CC) and TET12SN(3CC) was comparable to the design based on fully orthogonal building modules with efficient refolding properties (Fig. 4j, k). In contrast, TET12SN(3CC)-neg exhibited substantial aggregation, suggesting formation of misfolded species in agreement with predicted suboptimal folding by the model (Fig. 4l).

Solution SAXS measurements were used to examine the effect of CC repeats on the structure (Fig. 5). Scattering profiles for variants TET12SN(2CC), TET12SN(22CC) and TET12SN (222CC) (Fig. 5a–c) were nearly identical with radius of gyration ($R_g$) of $3.43 \pm 0.01$ nm. The scattering profiles fit well to CCPO design models with chi value ($\chi$) between 1.0 and 1.5 (Fig. 5a–f). Accordingly, the maximum particle distance ($D_{max}$) determined from the distribution of pair distances $P(r)$ was in the range of $10.0 \pm 0.5$ nm for all cage designs (Fig. 5g). Ab initio reconstruction led to tetrahedral molecular envelopes with a clearly discernible central cavity (Fig. 5d–f). Kratky plot signalled the presence of multiple domains connected with a flexible linker, which is consistent with the CCPO design, since CC modules representing cage edges are connected via a pentapeptide GSGPG linker (Fig. 5h). To quantitatively compare the measured scattering profiles of different designs, we calculated their SAXS similarity matrix (Fig. 5i)[34]. SAXS data for TET12SN and theoretical scattering profile calculated for an ideal tetrahedral structure[10] were also included in the calculation. Scattering curves of CCPO tetrahedron variants with CC repeats were indeed highly similar with a volatility of ratio ($V_R$) for all pairwise comparisons below 2.0. Interestingly, the variants containing repetitions of CC modules were even closer to the ideal tetrahedral structure[10] than TET12SN, with TET12SN(22CC) being closest to the ideal shape.

SAXS profiles of 3CC variants were on the other hand markedly different in comparison to the 2CC series of CCPO tetrahedrons with higher $R_g$ values and the absence of a maximum at $q = 0.15\,Å^{-1}$ (Supplementary Fig. 16). SAXS similarity matrix showed that they fit poorly to a tetrahedral cage structure (Supplementary Fig. 17). To confirm the observed misfolding of the structure was due to the high number of identical CC repetitions and not due to sequence specific effects two more variants were tested, where the CC pair P3SN:P4SN instead of P5SN:P6SN was repeated three times (Supplementary Fig. 18, Supplementary Table 10). The designed variants were even more poorly behaved and we were unable to obtain them in pure monomeric form due to a large amount of dimeric and trimeric species that could not be removed by SEC (Supplementary Fig. 18). Furthermore, a significant amount of degradation

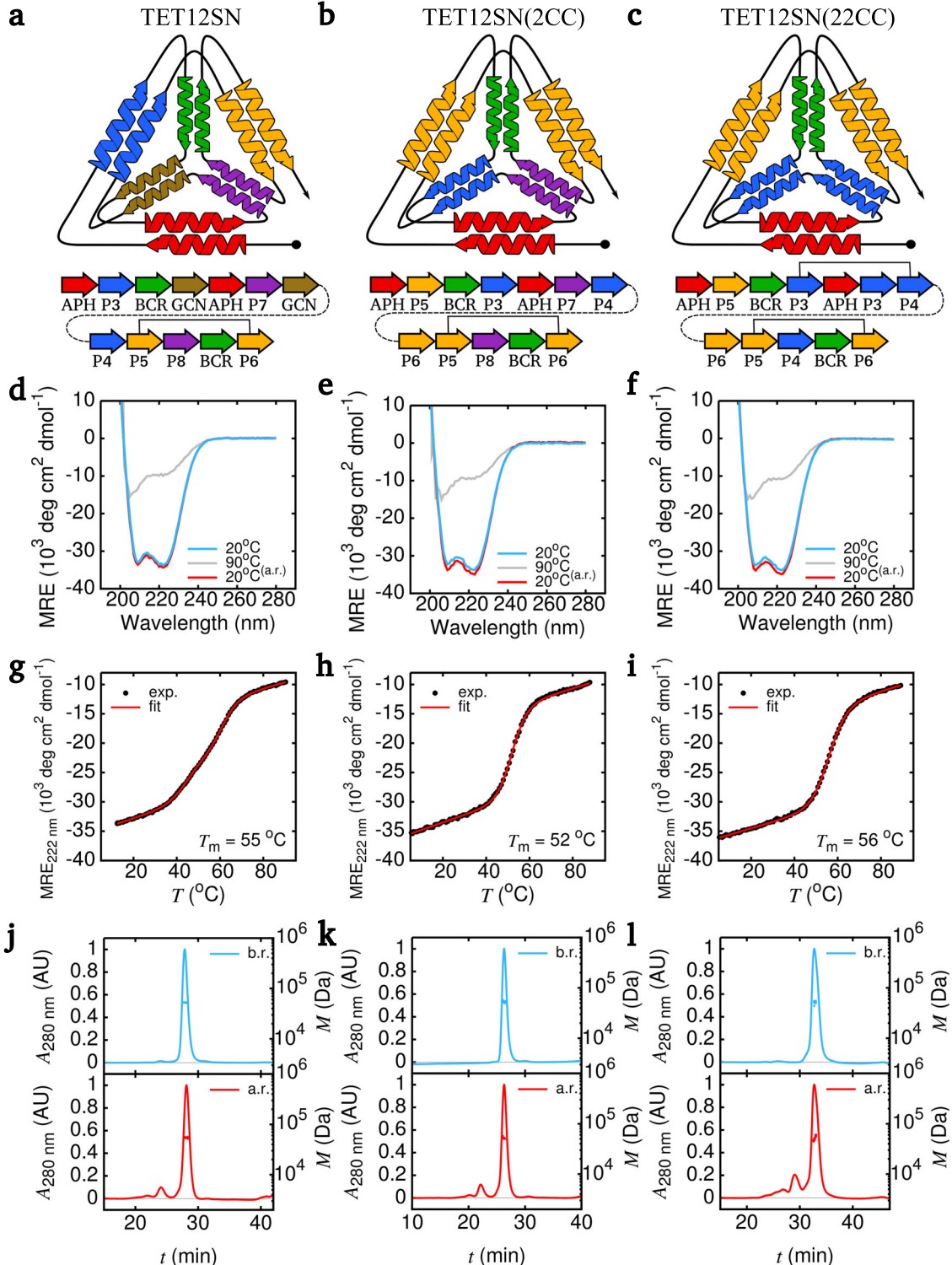

**Fig. 3 Biophysical characterisation of designed tetrahedral protein cages with one or two CC building modules repeated in the amino acid sequence.** Schematic illustration of designed CCPO cages. TET12SN contains only unique CC modules (**a**), TET12SN(2CC) contains two copies of the coiled-coil pair P5SN:P6SN (**b**), while two copies of both P3SN:P4SN and P5SN:P6SN are present in TET12SN(22CC) (**c**). In all cases, peptide modules are connected by a GSGPG linker. For reoccurring CC building blocks, the solid black line is intended to help discern on-target peptide pairings. **d–f** Circular dichroism (CD) spectra at 20 °C (blue), 90 °C (grey) and 20 °C after denaturation and rapid refolding (red). CD signal was converted to mean residue ellipticity (MRE) to facilitate comparison. **g–i** Temperature denaturation was monitored with CD at 222 nm. Melting temperatures ($T_m$) were determined using a two-state model. **j–l** Monodispersity of protein cage solution samples was analysed by SEC coupled to MALS before (blue) and after (red) temperature denaturation. Elution profiles were monitored by UV absorbance at 280 nm ($A_{280\ nm}$). Molecular weight ($M$) was determined from light scattering. Source data are provided as a Source Data file.

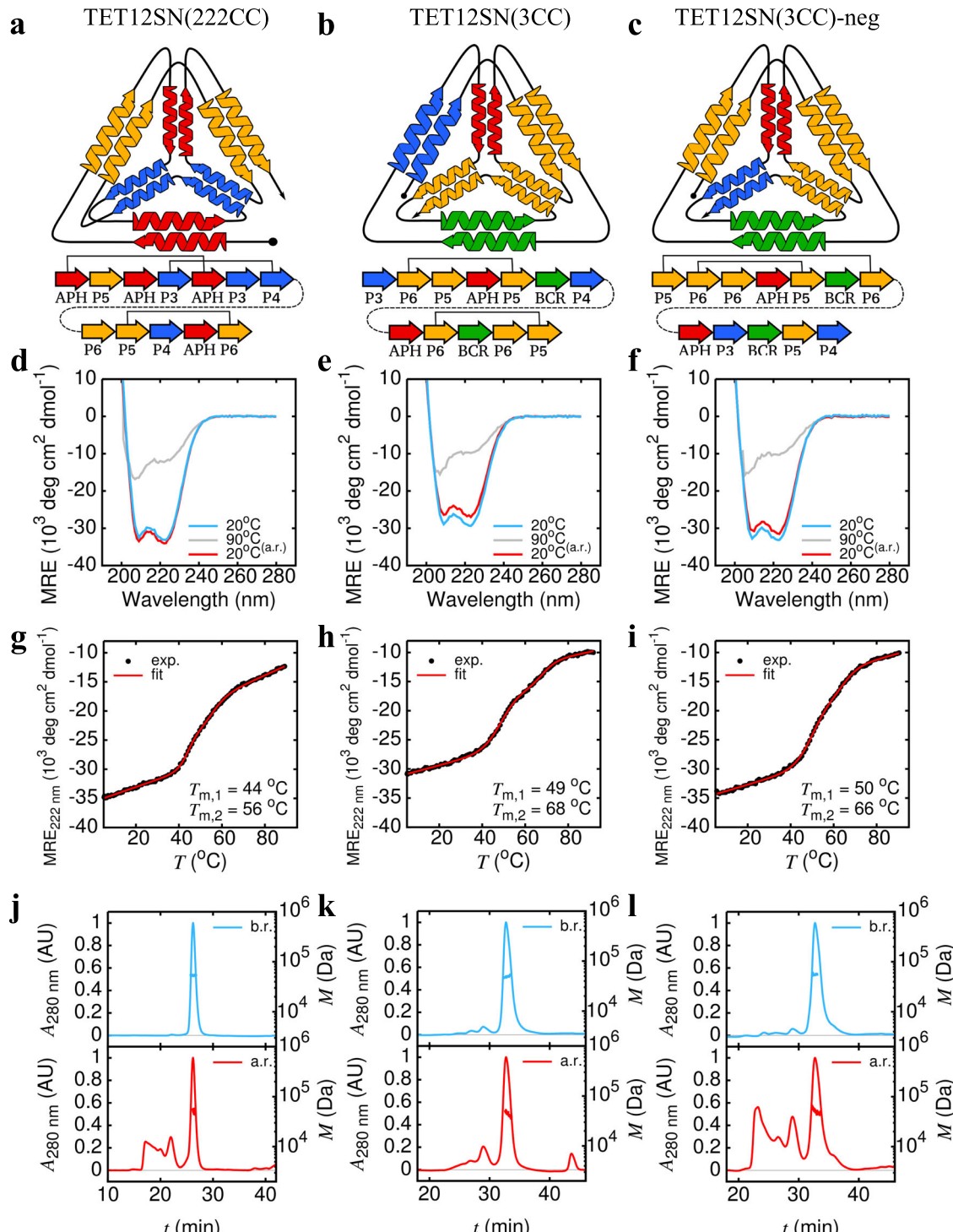

**Fig. 4 Biophysical characterisation of tetrahedral protein cages with three duplications or a single triplication of the same CC building module.**
Schematic depiction of designed CCPO cages. TET12SN(222CC) (**a**) comprises two copies of coiled-coil pairs P5SN:P6SN, P3SN:P4SN and APHSN.
TET12SN(3CC) (**b**) and TET12SN(3CC)-neg (**c**) on the other hand contain three instances of P5SN:P6SN in their amino acid sequence. For repeating CC
modules, on-target peptide pairings are indicated with a solid black line. **d–f** Circular dichroism (CD) spectra at 20 °C (blue), 90 °C (grey) and 20 °C after
denaturation and rapid refolding (red). CD signal was converted to mean residue ellipticity (MRE) to facilitate comparison. **g–i** Temperature denaturation
was monitored with CD at 222 nm. In all cases, the denaturation curve was characterised by two transitions. Melting temperatures associated with each
transition ($T_{m,1}$ and $T_{m,2}$) were determined using a three-state model. **j–l** SEC-MALS analysis before (blue) and after (red) temperature denaturation
revealed a high degree of aggregation after refolding for TET12SN(3CC)-neg. Elution profiles were monitored by UV absorbance at 280 nm ($A_{280\ nm}$).
Molecular weight ($M$) was determined from light scattering. Source data are provided as a Source Data file.

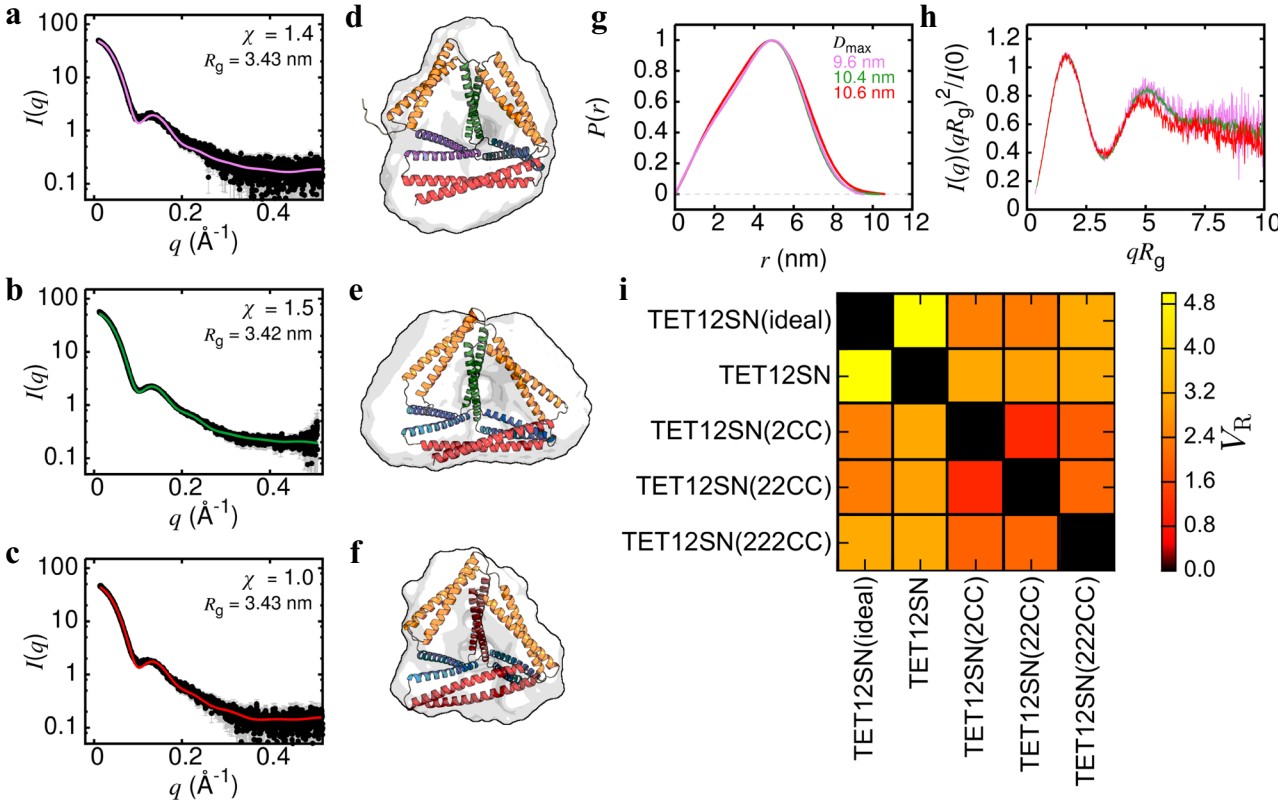

**Fig. 5 Solution SAXS analysis of CCPO cage structures comprising multiple repeats of CC building modules.** Scattering intensities $I(q)$ as a function of the scattering vector $q$ for (**a**) TET12SN(2CC), (**b**) TET12SN(22CC) and (**c**) TET12SN(222CC). Designs were characterized by a highly similar radius of gyration ($R_g$). SAXS data (black dots) exhibited good agreement with the theoretical scattering curves (solid lines) calculated from molecular models of designed CCPO cages shown in (**d–f**). The agreement was evaluated using the chi ($\chi$) metric. SAXS data are presented as mean values ± SD ($n = 40$ technical repetitions). **d–f** Ab initio reconstruction led to tetrahedral molecular envelopes with a clearly visible central cavity. **g** The distribution of pair distances $P(r)$ and the maximum particle distance ($D_{max}$) were comparable for all CCPO cage designs. The pink trace corresponds to TET12SN(2CC), green to TET12SN(22CC), while the $P(r)$ for TET12SN(222CC) is shown in red. **h** Kratky plots reflected the presence of multiple domains connected with a flexible linker, in accordance with the design strategy. Colour coding as in (**g**). **i** SAXS similarity matrix showing quantitative comparison between SAXS data obtained for different cage designs and theoretical scattering calculated for an idealised tetrahedral cage[10]. Similarity was evaluated using the volatility of ratio metric ($V_R$). Source data are provided as a Source Data file.

was observed. This suggests that the designed variants folded poorly already during production in bacteria.

## Discussion

Historically, folding has been described as a stepwise process including several partially structured intermediates. This view was upended in the '90s, when experimental studies indicated many natural proteins exhibit a single cooperative folding transition[35]. Subsequent advances in experimental techniques turned the tables once more, indicating proteins proceed to the folded state in several steps, sampling several pathways along the way[35,36], leading to an emergence of a more balanced view, which proposes that both folding models represent extreme examples of a common underlying mechanism. Still, identifying multi-step folding is experimentally challenging, since multi-step folders can exhibit an apparent two-state transition if the energy barriers between structural intermediates are low. While asynchronous structural changes have been hitherto observed for several proteins[36,37], the generalizability of reported experimental outcomes is hard to discern, since most studies focused on an individual model protein. In addition, folding studies were almost exclusively concerned with folding of single protein domains, while the assembly of larger multi-domain proteins remains largely unexamined.

Due to their particular topological architecture, CCPO cages are an interesting model system for studying folding of modular proteins. In contrast to natural proteins, where the fold is defined by a large number of weak cooperative interactions, modular protein folds are defined by more pronounced hierarchical interactions between discrete folding units. Due to the highly modular nature of CCPO assemblies, the relative contribution of the intra-chain distance between CC peptide pairs to folding in opposition to their enthalpic stability can be estimated. Molecular simulations and multi-site FRET experiments revealed that folding of CCPO cages proceeded sequentially, with a single CC module forming in each step. We demonstrated that the intra-chain distance between interacting folding units, which affects their local concentration, dominates the folding of CCPO cages, and trumps CC units' intrinsic thermodynamic or kinetic stability. However, while our results point to the dominant role of chain topology in determining the folding pathway, additional factors likely play a role as well. For example, while APHSN is the first to fold in TET12SN, three CC modules share the same intra-chain distance (Fig. 2c). While this might be due to its higher thermodynamic stability, it remains to be examined. Regarding the formation of secondary structure, equilibrium stability measurements suggested helices partially form before the assembly of CC modules, however the main increase in helical content occurred concomitantly with CC formation.

Nevertheless, the robust underlying modularity of CCPO cages, with intra-chain distance easily defined for each CC pair, enabled

the construction of a radically simple folding model that takes into account only the intra-chain distance between the CC-forming peptides. The distance-dependent model was utilized to design several CCPO tetrahedra containing multiple copies of the same building block, demonstrating that the construction of CCPO cages does not require unique orthogonal building blocks. Increasing the number of CC modules that repeat two times in the amino acid sequence maintained the fold and efficient refolding of the tetrahedral protein cage. Furthermore, SAXS comparison demonstrated the structure of CCPO tetrahedron variants with one or more CC modules occurring two times in the amino acid sequence resembled the ideal tetrahedral shape more closely than that of TET12SN. The latter might be due to the elimination of the GCNSN pair, which is slightly shorter than other building blocks, resulting in a more symmetric cage structure.

In accordance with predictions from the folding model, correct positioning of repeating CC modules appears to be crucial. While the design of CCPO cages with multiple occurrences of the same peptide segments were based on the preferential interaction between nearest complementary segments, negative design was utilized as well, taking advantage of sterical hindrances and the defined orientation of CC dimers (e.g., parallel coiled-coil forming peptides cannot assemble if positioned adjacently). The importance of segment positioning is most clearly seen when comparing in vitro refolding of the positive and negative design variants of TET12SN(3CC), where the latter exhibited a much higher degree of aggregation upon refolding than the positive design variant, where the arrangement of segments was optimal, despite both designs having the same predicted thermodynamic stability. Formation of aggregates upon refolding likely occurred due to misfolding resulting in exposed non-paired segments, which are prone to aggregation.

In summary, successful construction of CCPO tetrahedra containing several identical CC building block repeats confirmed our distance-dependent folding model. The proposed mathematical folding model was also successfully applied to the design of a larger bipyramidal CCPO cage containing multiple instances of several CC building modules, highlighting its generalizability[38]. By reusing CC modules in the polypeptide sequence, the number of unique CCs required for constructing a CCPO cage can be decreased by up to 50%. Relaxation of the requirement for orthogonal sets of parallel and antiparallel CC dimers importantly increases the complexity of CCPO cages that can be achieved with an available orthogonal coiled-coil set. Currently, the orthogonal CC building block set used for CCPO construction consists of ~10 CC modules, encompassing both natural and synthetic CCs in parallel or antiparallel orientation, which can be used simultaneously, limiting designability to shapes composed of maximally 10 edges. Using each module twice in CCPO design will allow construction of larger, more complex polyhedral cages with up to 20 edges, such as octahedron, pentagonal bipyramid, snub disphenoid or elongated triangular bipyramid. The strategy of recycling building modules could likely be applied also for DNA nanostructures, although in the case of polynucleic acids the number of orthogonal building modules is typically not the limiting step. On the other hand, the design of the folding pathway has been utilized to introduce knot formation in single chain DNA nanostructures[15], where application of simple design rules enabled construction of efficiently folding structures.

Our findings regarding the importance of distance between intra-chain native contacts for the folding pathway also bear relevance for understanding the folding of naturally occurring tandem repeat proteins. The latter are often composed of highly similar domains such as in the case of the Ankyrin repeat family[39]. In natural multi-domain proteins, homologous domains

are rarely positioned adjacently in the amino acid sequence. Our results support the hypothesis that the latter serves to prevent formation of stable misfolded states[21,25,40]. Avoiding misfolding is of paramount importance, since it not only competes with productive folding, resulting in non-functional proteins, but can also lead to formation of large-scale aggregates connected to various debilitating illnesses[41]. Whether our results could be generalised to proteins that contain less clearly defined folding units remains to be seen.

## Methods

**Materials**. The peptides with their N-termini protected by acetylation and C-termini protected by amidation were purchased from Proteogenix (France). Synthetic genes were purchased from Twist Bioscience (CA, USA). All other chemicals were purchased from Sigma-Aldrich.

**Gō simulations**. All-atom (except hydrogens) folding simulations were performed using the Gō force field[32] and GROMACS simulation package[42]. Topology files were created with the SMOG webserver[43] based on CCPO model structures confirmed by SAXS[10]. Native contacts were calculated using the Shadow algorithm[44]. Atoms were assigned zero charge and masses in accordance with their atomic mass. To allow for a longer time step ($t^* = 0.002$), the strengths of bond angles were increased twofold, while the bonds were weakened by the same factor. The simulations were performed in vacuum without periodic boundary conditions. Each CCPO cage was quickly unfolded by simulating molecular dynamics at high temperature (GROMACS temperature $T = 200$), snapshots of the unfolded structure were used as the starting point for folding simulations (100 runs, GROMACS temperature $T = 100$, 30 million steps). The obtained trajectories were analysed to determine the order of CC formation. A CC was considered formed once 50% or more of its native contacts were reconstituted.

**Design of CCPO tetrahedron cages**. The tetrahedral protein cage with edges represented by CC dimers can be constructed from three different topologies[10]. As a starting point for the design, we chose the so-called Topology 1 that has been hitherto best experimentally characterised. For each design case containing a particular number of CC repetitions, the best arrangement of CC building blocks was found by predicting the probability of folding, $P_F$, for all possible circular permutations and peptide segment arrangements associated with Topology 1, using the deterministic model of folding as described in the main text. Molecular models of CCPO tetrahedra with arrangements yielding the highest $P_F$ were built using the CoCoPOD computational platform for automatic CCPO cage modelling, available at https://github.com/NIC-SBI/CC_protein_origami[10].

**Molecular cloning**. TET12SN(22CC), TET12SN(3CC) and TET12SN(3CC)-neg were ordered as synthetic genes in pET-41a(+) from Twist Bioscience (CA, USA) (Supplementary Table 11). All other genes were constructed by Golden Gate Assembly (GGA)[45,46] or by Gibson assembly (GA). For GGA, each sequence coding for a tetrahedral construct was divided into 12 fragments, each corresponding to one CC peptide (Supplementary Table 12). Custom Golden Gate overhangs, flanking CC fragments on both sides, were designed in such a way, that upon digestion with BsaI, overhangs enabled hierarchical and scarless assembly of basic building blocks into final constructs. For GA, amplification of the vectors and DNA inserts (primers in Supplementary Table 13) was performed with Phusion® HotStart DNA polymerase (NEB, MA USA) in PCR reactions carried out according to manufacturer instructions. Next, ligation was achieved by incubating up to 50 ng of DNA (insert and plasmid) for 1 h at 50 °C in the presence of a mixture of the following enzymes: Taq Ligase (NEB, MA USA), Phusion® Polymerase (NEB, MA USA) and T5 exonuclease (NEB, MA USA).

**Protein expression and purification**. Protein constructs prepared in pET-41a(+) were transformed into *Escherichia coli* NiCO21 (DE3) strain (NEB, MA, USA) and grown overnight at 37 °C on LB agar plates supplemented with kanamycin (50 μg/mL). Inoculums were prepared by picking single colonies and growing them overnight at 37 °C at 160 rpm in 100 mL of LB medium, supplemented with kanamycin (50 μg/mL). Inoculums of bacterial cultures were added to 5 L fermentation flasks, each containing 1.5 L of LB media, to reach OD level of 0.1 and grown at 37 °C. Upon the bacterial culture reaching OD 0.6, 0.5 mM IPTG (Goldbio, MO, USA) was added. After 4 h of additional growth at 30 °C, the bacterial pellets were collected by centrifugation. Bacterial pellets were stored at −80 °C overnight. Frozen pellets from 3 L of fermentation were resuspended on ice in 40 mL of lysis buffer (50 mM Tris at pH 8.0, 150 mM NaCl, 10 mM imidazole, 15 U/mL Benzonase (Merck, Germany)) and CPI protease inhibitor mix (Millex Sigma-Aldrich, MO, USA), and lysed by sonicating (intervals of 1 s ON, 3 s OFF, effective sonication per cycle 1 min, amplitude 55%) for 6 cycles or until the suspension clarified. Soluble fractions were obtained by centrifugation at 16000 × $g$ (4 °C) for 20 min, filtered through 0.45 μm syringe filter units (Sartorius stedim,

Germany) and added to 7 mL of previously equilibrated Ni-NTA resin (Goldbio, MO, USA) on gravity columns. Ni-NTA columns were washed with buffer A (50 mM Tris at pH 8.0, 150 mM NaCl, 10 mM imidazole) and B (50 mM Tris at pH 8.0, 150 mM NaCl, 20 mM imidazole). Bound fraction was eluted with careful addition of elution buffer (50 mM Tris at pH 8.0, 150 mM NaCl, 250 mM imidazole). Fractions containing the protein of interest were merged and injected into a size exclusion column (HiLoad 26/600 Superdex 200 pg, GE Healthcare, IL, USA) and separated at 2.6 mL/min (20 mM Tris at pH 7.5, 150 mM NaCl, 10 % (v/v) glycerol). Appropriate fractions were merged, concentrated (centrifugal unit 10 MWCO, Amicon-ultra, Millex Sigma-Aldrich, MO, USA), plunge frozen in liquid nitrogen and stored at −80 °C. The purity of isolated proteins was analysed with SDS-PAGE (for uncropped gels see Source Data).

**Fluorescent labelling.** Proteins were fluorescently labelled with Sulfo-cy3 and Sulfo-cy5 (Lumiprobe, MD, USA) via thiol-maleimide coupling. Sulfo-cy3 and Sulfo-cy5 were dissolved in DMSO, mixed together and added in 5-times molar excess to approximately 1 mL of 1 mg/mL protein solution in Tris buffer (20 mM Tris at pH 7.5, 150 mM NaCl) with 1 mM TCEP. The reaction was allowed to proceed over night at 4 °C. The reaction mixture was purified on a PD 10 disposable desalting column (GE Healthcare, IL, USA), using Tris buffer as mobile phase.

Peptide labelling reactions were performed by mixing 100 nmol of peptide dissolved in 100 μL of labelling buffer (50 mM Tris, 150 mM NaCl, 1 mM TCEP, 5 mM EDTA, pH = 7.2) with 300 μL of DMSO and adding 50 μL of 13 mM dye solution (in DMSO). The reaction was allowed to proceed for two hours at room temperature, protected from ambient light. The labelled peptide was isolated from the reaction mixture with reverse phase HPLC using a Jupiter 4 μm Proteo 90 Å column (Phenomenex, CA, USA). Separations were performed with two buffer systems. For P7SN-cy3 and APHSN-cy3 phase B (95% ACN, 5% $H_2O$, 0.1% TFA) was non-linearly increased from 0 to 80% in 26 min (phase A: 5% ACN, 95% $H_2O$, 0.1% TFA, Supplementary Table 14a). For other peptide-dye conjugates phase B (95% ACN, 5% $H_2O$) was increased from 0 to 80% in 36 min (phase A: 5 % ACN, 95% 0.1 M TEAA pH = 7, Supplementary Table 14b). After isolation peptides were exchanged into buffer (50 mM Tris, 150 mM NaCl, pH =7.5). Concentration was determined spectrophotometrically from dye absorbance ($\varepsilon_{548}^{Sulfo-cy3} = 162000\ \text{L mol}^{-1}\text{cm}^{-1}$, $\varepsilon_{646}^{Sulfo-cy5} = 271000\ \text{L mol}^{-1}\text{cm}^{-1}$). Mass and purity of the isolated peptide-dye conjugates were confirmed by matrix-assisted laser desorption/ionization time-of-flight mass spectrometry (MALDI-TOF).

**Matrix-assisted laser desorption/ionization time-of-flight mass spectrometry (MALDI-TOF).** Bruker UltrafleXtreme MALDI-TOF mass spectrometer (Bruker Daltonics, Bremen, Germany) was used to perform mass spectrometry experiments. Prior to the measurements, each peptide-dye sample was combined in 1:1 volumetric ratio with a saturated solution of sinapinic acid dissolved in acetonitrile and 0.1% trifluoroacetic acid (7:3, v/v). Using the dried-droplet method, 1 μL of the sample was then spotted on the target plate. The mass spectra of the peptide-dye samples were recorded in the reflective positive ion mode. External calibration was performed using peptide calibration standards and protein calibration standards I (Bruker Daltonics).

**Förster resonance energy transfer (FRET) stopped-flow.** Labelled CCPO cages were unfolded by adding 478 mg of guanidine hydrochloride (Gdn-HCl) into 1 mL of protein samples at a concentration of ~1 mg/mL resulting in Gdn-HCl final concentration of ~5 M. Coiled-coil forming peptide-dye conjugates were mixed in 1:1 ratio resulting in 20 μM dimer concentration. Subsequently, Gdn-HCl was added to solutions to achieve a final concentration of 5 M. The samples were left for one hour at room temperature to ensure complete unfolding.

The measurements were performed on MOS-200 spectrometer with an SFM-3000 stopped flow unit (BioLogic, France). Data was collected with Bio-Kine32 software (version 4.72.). Total flow was set to 6 mL/s (estimated dead time 3.7 ms) or 11.5 mL/s (dead time 1.9 ms). Folding of CCPO cages was triggered by mixing at least 30 μL of sample against 4 volumes of Tris buffer (20 mM Tris, 150 mM NaCl, pH = 7.5) at different Gdn-HCl concentrations (0–2.5 M) resulting in postmix Gdn-HCl concentrations between 1 M and 3 M. For refolding experiments concerning individual CC dimers the measurements were performed at 1 M postmix denaturant concentration. The cuvette used was 1.5 mm by 0.5 mm in size (FC15/5). The instrument was set in fluorescence mode with the detector at 90° to the excitation light. The sample was excited at 547 nm. Fluorescence above 655 nm was recorded each ms for 3 s. Obtained stopped-flow traces were averaged (n ≥ 3) and fit to a two-state kinetic model:

$$y = a\exp(-kt) + bt + c, \tag{1}$$

where $y$ is FRET signal, $t$ time, $k$ folding constant, $a$ FRET amplitude, $b$ drift parameter and $c$ is background noise. The fitting was carried out in Python (version 3.5.4) using the lmfit module (version 0.9.11).

**Fluorescence spectroscopy.** Fluorescence spectra of fluorescently labelled TET12SN variants were recorded with a multi-plate fluorescence reader Synergy Mx using the Gen5 program (version 1.10, Bio TeK, VT USA). Following excitation

at 528 nm the fluorescence was measured between 548 and 800 nm. FRET intensity was calculated as the ratio between acceptor emission at 668 nm and donor emission at 566 nm. All measurements were performed in triplicates. FRET intensities as a function of Gdn-HCl concentration were described using a two-state model

$$\text{FRET} = B + \frac{C(1 + D[\text{Gdn} - \text{HCl}])}{1 + \exp\left(-\frac{1}{RT}(\Delta G_{\text{fold}} - m[\text{Gdn} - \text{HCl}])\right)}, \tag{2}$$

where $\Delta G_{\text{fold}}$ is the free energy of unfolding, $T$ temperature, $R$ gas constant, $m$ rate of change of $\Delta G_{\text{fold}}$ with Gdn-HCl concentration and $B$, $C$ and $D$ are fitting parameters.

Change in acceptor emission during thermal denaturation was monitored using LightCycler 480 Real-Time PCR instrument (Roche, Germany; software version 1.5.1.62) using 523 nm/670 nm filter combination. Melting temperatures were determined by fitting a thermodynamic model to experimental data (see Eqs. (4)–(11))[47].

**Circular dichroism (CD) spectroscopy.** CD spectra and thermal denaturation curves were recorded on a ChiraScan instrument (Applied Photophysics, UK; software version 4.5), equipped with a Peltier thermal control block (Melcor, NJ, USA, now part of Laird Technologies). CD spectra were recorded in a 1 mm quartz cuvette from 200 to 280 nm with a 1 nm step size, bandwidth of 1 nm and 0.5 s integration time. Spectra and denaturation curves of individual CCs were measured at 40 μM dimer concentration. All measurements were performed in three technical replicates and subsequently averaged. The helical content was derived from the following equation[48]:

$$\text{Helical content}\ (\%) = \text{MRE}_{222} / \left(\text{MRE}_{222}^{\text{H}} \times (1 - 2.57/n)\right), \tag{3}$$

where $n$ is the length of amino acid sequence, $\text{MRE}_{222}$ average mean residue ellipticity at 222 nm, and $\text{MRE}_{222}^{\text{H}}$ is the theoretical mean residue ellipticity of an infinitely long helix ($-39{,}500\ \text{deg cm}^2\ \text{dmol}^{-1}$)[48].

Temperature denaturation experiments were conducted by heating samples from 0 to 90 °C at a rate of 1 °C/min. CD signal was measured at 222 nm at 1 °C increments. The effective sample temperature was monitored through a temperature probe inserted in the cuvette. Melting temperatures were determined by fitting a thermodynamic model to experimental CD data using the glox software (version 0.2.)[47]. For two-state denaturations, the fraction of the protein in the folded state $\alpha_f^{\text{model}}$ was calculated as:

$$\alpha_f^{\text{model}}(T) = \frac{\exp\left(-\frac{1}{RT}\left(\Delta H\left(1 - \frac{T}{T_m}\right)\right)\right)}{1 + \exp\left(-\frac{1}{RT}\left(\Delta H\left(1 - \frac{T}{T_m}\right)\right)\right)}, \tag{4}$$

where $R$ is the gas constant, $T$ temperature, $\Delta H$ folding enthalpy and $T_m$ melting temperature. $\Delta H$ and $T_m$ were optimized to minimize the difference between $\alpha_f^{\text{model}}$ and the experimentally observed fraction of the protein in the folded state $\alpha_f^{\text{exp}}$ expressed as:

$$\alpha_f^{\text{exp}}(T) = \frac{\text{MRE}_{222}(T) - \text{MRE}_{222}^u(T)}{\text{MRE}_{222}^f(T) - \text{MRE}_{222}^u(T)}. \tag{5}$$

$\text{MRE}_{222}^u(T)$ and $\text{MRE}_{222}^f(T)$ correspond to the MRE at 222 nm expected for the unfolded and the folded state, respectively, and are estimated from the unfolded and folded state baselines.

In case the denaturation proceeded as a three-state process, the following set of equations was used:

$$U \overset{K_i}{\leftrightarrow} I \overset{K_f}{\leftrightarrow} F, \tag{6}$$

$$\alpha_f^{\text{exp}}(T) + 0.5\alpha_i^{\text{exp}}(T) = \frac{\text{MRE}_{222}(T) - \text{MRE}_{222}^u(T)}{\text{MRE}_{222}^f(T) - \text{MRE}_{222}^u(T)}, \tag{7}$$

$$\alpha_i^{\text{model}}(T) = \frac{K_i}{1 + K_f K_i + K_i}, \tag{8}$$

$$\alpha_f^{\text{model}}(T) = \frac{K_f K_i}{1 + K_f K_i + K_i}, \tag{9}$$

$$K_i = \exp\left(-\frac{1}{RT}\left(\Delta H_i\left(1 - \frac{T}{T_{m,1}}\right)\right)\right), \tag{10}$$

$$K_f = \exp\left(-\frac{1}{RT}\left(\Delta H_f\left(1 - \frac{T}{T_{m,2}}\right)\right)\right), \tag{11}$$

where subscript i denotes the intermediate state.

**Size exclusion chromatography coupled to multi-angle light scattering (SEC-MALS).** SEC-MALS experiments were performed on a Waters e2695 HPLC system coupled with a 2489 UV detector (Waters, MA, USA), a Dawn8+ multiple-angle

light scattering detector (Wyatt, CA, USA) and refractive index (RI) detector RI500 (Shodex, Japan). Protein samples were filtered using Durapore 0.1 μm centrifuge filters (Merck Millipore, MA, USA) after which 100 μL were injected onto a Superdex 200 Increase 10/300 GL column (GE Healthcare, IL, USA). Samples were eluted in 20 mM Tris, 150 mM NaCl, pH = 7.5 at a flow rate of 0.5 mL/min. Data analysis was carried out using Astra 7.0 software (Wyatt, CA, USA) utilizing the RI signal as the concentration source.

**Small angle X-ray scattering (SAXS)**. SAXS experiments were performed at the P12 beamline[49] at the PETRA-III synchrotron (DESY, Hamburg, Germany). The wavelength of the incident X-ray beams was 1.24 Å. Scattering was recorded with the Pilatus 6 M detector, which was placed 3 m from the sample, resulting in the scattering vector ranging from 0.028 to 7.3 nm$^{-1}$. The measurements were performed in flow-through mode using the robotic sample handler[50]. For each measurement, 30–40 μL of sample was used. Scattering was gathered over 40 frames lasting 0.05 or 0.1 s. The frames affected by radiation damage were discarded and the remaining frames automatically averaged[51]. In addition, buffer scattering was measured before and after each protein sample and subsequently subtracted from the sample scattering. For each protein sample, 4 concentrations were measured (~8, 4, 2 and 1 mg/mL) to evaluate concentration effects. Merging and subsequent analysis was performed using ATSAS[52,53] software. P3 symmetry was imposed during ab-initio model generation. Theoretical scattering curves were calculated and compared to experimental SAXS profiles using PepsiSAXS (version 3.0)[54]. SAXS similarity matrix was calculated using the on-line SAXS heat map application at https://sibyls.als.lbl.gov/saxs_similarity[55].

**Reporting summary**. Further information on research design is available in the Nature Research Reporting Summary linked to this article.

## Data availability
SAXS data for TET12SN(2CC), TET12SN(22CC) and TET12SN(222CC) have been deposited to SASBDB (accession codes: SASDKQ2, SASDKR2 and SASDKS2, respectively). Source data are provided with this paper.

## Code availability
The code for predicting the probability of folding for CCPO cages with repeating CC building blocks is provided as Supplementary Data. The code is distributed under the MIT License.

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

## Acknowledgements
This project was financed by Slovenian Research Agency (program no. P4-0176), ERANET project Medisurf and by the ERC AdG project MaCChines to R.J. This work has been supported by iNEXT, grant number 653706, funded by the Horizon 2020 programme of the European Commission (PID: 2437, 2706). The synchrotron SAXS data was collected at beamline P12 operated by EMBL Hamburg at the PETRA III storage ring (DESY, Hamburg, Germany). We would like to thank Stefano Da Vela for the assistance in using the beamline. We would like to thank Bojana Stevovič for help in experimental work.

## Author contributions
J.A., I.D., and A.L. wrote the code for analysing MD folding trajectories. J.A. performed and analysed MD simulations. J.A., Ž.S., and F.L. performed the stopped-flow folding experiments. D.P. conducted MALDI-TOF experiments. J.A. and T.P. devised the deterministic folding model. J.A. designed CCPO tetrahedron variants. Ž.S. cloned and purified the proteins and performed biophysical characterisation. F.L. contributed to protein purification. J.A. performed the SAXS experiments and SAXS data analysis. J.A., Ž.S, F.L., and R.J. designed the study. J.A. and R.J. wrote the initial manuscript. All authors discussed the results and contributed to the manuscript.

## Competing interests
The authors declare no competing interests.
