## [Peer Review File · Nature Communications]

REVIEWER COMMENTS

Reviewer #1 (Remarks to the Author):

In this article, the authors demonstrate that the designed coiled-coil protein origami (CCPO) folds in a stepwise sequential pathway. MD simulations and stopped-flow FRET measurements revealed that CCPO folding is dominated by the effective distance between CC modules in the primary sequence and subsequent folding intermediates. The number of orthogonal modules required for constructing a CCPO tetrahedron could be reduced from six to as little as three different CC modules. There are some interesting results with originality in the manuscript. However, I think some explanations are insufficient and it is difficult for a broad audience to understand results and discussion thoroughly. Overall, I cannot recommend this manuscript to publish in a high-quality multidisciplinary journal, Nature Communications, unless extensive revisions are made.

To improve this manuscript, I would like to give the authors several suggestions as follows.

- (1) What is the definition of "intra-chain distance"? How did you count it? Please give clear and detailed explanations for a broad audience.
- (2) Figure 1 is too complicated to follow. Please divide it into some figures and a table in an easy-to-understand manner. In addition, it is difficult for me to understand the graph in the lower panel of Figure 1a. What are the definitions of "contact order" and "effective distance" in this manuscript? How did you calculate them? Please give clear and detailed explanations.
- (3) In Figure 2, the authors should show the results of TET12SN as a control.
- (4) The authors should add some movies of typical results of MD simulations in supplementary information.
- (5) Please add page numbers in the manuscript.
- (6) In the lower panel of Supplementary Figure 1, how did you figure out which pair was observed to be first to fold?

Reviewer #2 (Remarks to the Author):

The authors provide an interesting set of experimental and computational data, investigating the folding pathway for a designed coiled-coil tetrahedral protein. They develop a model for the folding pathway based only on the inter-chain distance between complementary coils, and use this to design simplified sequences, which are still able to fold into the desired structures, as determined by SAXS. I think the greatest strength of the paper is the use of the developed model in the successful design of the simplified sequences. However, the work could be greatly strengthened with the addition of some key experiments / controls which would allow the authors to more fully substantiate their folding model. I think this is interesting work and may be publishable in Nature Communications after the authors address the following questions / comments:

- 1) It would be really useful to see some additional experimental investigation of the proposed folding pathway. This would help to convince readers of the sequential nature of the folding process. Additionally some more detailed discussion of the folding process itself, with respect to the initial formation of helices, and their docking, would be helpful: e.g. it was not clear to me if the authors are proposing a classical framework model in which all the alpha helices form first, and then the closest complementary helices dock, or if the helices form cooperatively upon docking, but do this in sequence. This distinction between secondary structure formation and docking of local helices is important and needs further comment in the main text.

I list here some experiments which would provide useful information, though I understand in the current Covid climate that some / none of these may be possible to perform at this time, in which case some discussion of the current limitations of the work should be included in the main text.

- a) An equilibrium chemical denaturation curve followed by CD, FRET or intrinsic fluorescence (if possible) and preferably all three, or reference to these if it is already in the literature. This may allow the authors to detect equilibrium (un)folded intermediates not populated under thermal

denaturation. Also, it is crucial to show that the protein is fully folded in 2 M Gdn-HCl, the conditions for much of the kinetic refolding studies.

b) Ideally the thermal denaturation should be repeated with both heating and cooling cycles, to show the complete thermodynamic reversibility of this process.

c) Stopped-flow CD to monitor the un/refolding process might also yield insights into the sequential model, as refolding when followed by CD (rather than by FRET) may be a multi-state process, especially as two distinct transitions are seen in the thermal denaturation of some designed proteins (e.g. Fig 3).

Minor points from the text:

2) Consider adding an introductory figure with the protein structure and nomenclature as a main text figure 1 – rather than the first figure call out being to a supplementary figure.

3) The intra-distance measurement needs to be more clearly defined. How is this calculation done?

4) Fig S5a is somewhat confusing. It also seems to imply formation of the helices occurs concurrently with their docking, but this is not discussed (see 1 above).

5) Fig 2a,b: Please define the black lines linking various helices in the schematics under the structures.

6) Fig 2g,h: It is difficult to see the refolded (red) trace behind the blue. Consider offsetting?

7) The appearance of two transitions in the thermal denaturations in Figure 3 is interesting and worthy of discussion: what is the interpretation of this, compared for the single transitions present in Fig 2, and why does this difference occur?

8) The discussion starts by implying that historically protein folding was thought to be a two-state, cooperative process. This was in fact a new discovery in the 1990's by Jackson, Fersht and others. Rather the historical perspective assumed a multistate process, with hierarchical (framework) models, and hydrophobic collapse as the predominant theories. Nucleation-condensation, two-state mechanisms, and folding funnels were all late to the game! Please revise accordingly.

Timothy D Craggs
University of Sheffield

Reviewer #3 (Remarks to the Author):

The ability to design synthetic peptide sequences whose interactions and therefore final quaternary structures are predictable is a dream of protein nanoscience. The concept that this could be done using relatively simple interaction rules, somewhat akin to those applied in the DNA origami technique is very interesting and is something the Jerala group have pioneered using coiled-coil interactions. In this work the authors have attempted to understand the folding pathway of such protein origami designs and shown that it proceeds in a stepwise manner with interacting domains that are closest to each other pairing first. This is an original insight and with this knowledge they are also able to show that more than one pair of identical interacting domains can be employed in the same structure so long as the distance rules are adhered to. These results provide useful information for protein designers who may be able to use these findings in their own designs. Furthermore it may provide some insights into how natural multi-domain proteins fold.

Overall the work is of high quality and clearly presented. However we have identified a number of points both major and minor that require correction or improvement.

Major points

-Page 4, The G \bar{o} model is mentioned. For non-experts (like me) there is insufficient explanation of what this model is or how it works. A cursory explanation in the text would be very useful, particularly given the general readership of Nat. Comm.

-When introducing the concept of interchain distance it should be made clear that whether the two coiled-coil modules are parallel or antiparallel makes a difference to this calculation. At first this is not obvious when looking at figures

-Page 6 "Additionally, we observed that coiled-coil forming peptides positioned at either terminal end exhibited a slightly increased folding speed". This phrase is unclear and requires further explanation. Do the authors mean that if we compare two folding pairs with the same distance between them but one is closer to the terminus than the other, then the closer one folds quicker? This seems to be suggested from close inspection of Supp. Fig.1 but it should be explained more clearly and better indicated in the figures exactly which pairs demonstrate this point.

-Figure 1 Legend "In each step, the peptide pair with the shortest inter-pair distance forms". But in the figure there are 3 possible pairs that have a contact order of 3 (APHSN, GCNSN and P5SN:P6SN). Why is one shown happening first in preference to the others. Are there in fact three pathways? If so this should be indicated and explained.

-Figure 1 d) Differences in melting temperatures are shown. Could the authors include an explanation for the observed differences?

-Page 13: It is mentioned that TET12SN(3CC)-neg exhibited substantial aggregation in refolding in line with the suboptimal folding as predicted by the model. Can the authors explain then why, if this arrangement had the lowest PF it is not surprising that it was normally expressed, purified and folded before unfolding experiments took place? It actually looks reasonable on SDS PAGE in Supp. Fig. 7 and SEC (Supp. Fig. 10)

-Page 20: "Relaxation of the requirement for orthogonal sets of parallel and antiparallel CC dimers importantly increases the complexity of CCPO cages that can be achieved with an available orthogonal coiled-coil set." Perhaps the authors could give an example of what structures one is presumably now able to design given the available CC domains and the 50 % decrease of unique domains needed?

-All of the work presented has been done with a tetrahedron shape. Can the authors include data to show whether or not this applies to other shapes?

-CC dimers are insufficiently introduced and explained. Notably peptide synthesis methods are completely lacking

-Supplementary Figure 5: Here the authors introduced additional dotted CC to the scheme which appears to be in the case of blue either P3SN or P4SN and in the case of yellow P5SN and P6SN. Presumably this is meant to show the identical CC copies but this is not explained anywhere. Please explain in the legend

Minor points

-Figure 1 a, Colour of the brown arrow and bar does not match

-Figure 1a, figure legend. Change "inter-pair" to "intra-chain" as in the rest of the text

-On page 4, the authors refer to "contact order". Are contact order, intra chain distance and inter peptide distance all describing the same? The authors should define one and use throughout the text & figures. This would make the text easier to understand

-Page 13, line 2. Change "asses" to "assess"

- When folding times are mentioned it would be interesting to know the absolute folding times.

- Supp. Figure 1: What is the total number of runs of this simulation/ total number of occurrences? Please add this to the figure legend, so the reader does not need to look for it in the methods part.

-Supp. Table 1. Please define the meaning of the lower and upper case letters

REVIEWER COMMENTS

Reviewer #1 (Remarks to the Author):

In this article, the authors demonstrate that the designed coiled-coil protein origami (CCPO) folds in a stepwise sequential pathway. MD simulations and stopped-flow FRET measurements revealed that CCPO folding is dominated by the effective distance between CC modules in the primary sequence and subsequent folding intermediates. The number of orthogonal modules required for constructing a CCPO tetrahedron could be reduced from six to as little as three different CC modules. There are some interesting results with originality in the manuscript. However, I think some explanations are insufficient and it is difficult for a broad audience to understand results and discussion thoroughly. Overall, I cannot recommend this manuscript to publish in a high-quality multidisciplinary journal, Nature Communications, unless extensive revisions are made.

To improve this manuscript, I would like to give the authors several suggestions as follows.

(1) What is the definition of “intra-chain distance”? How did you count it? Please give clear and detailed explanations for a broad audience.

Due to the modularity of CCPO cages, we introduced the intra-chain distance as a simple metric for describing the changing proximity of pairing peptide segments during folding of CCPO cages. We defined intra-chain distance as the minimal number of peptide modules separating the termini of complementary peptide segment. For each CC module, two distances are obtained. For antiparallel CCs the distance is calculated between opposing termini (N-C), while for parallel CCs, matching termini are used for distance calculation (N-N and C-C). In both cases, the minimum of the two values is taken as the intra-chain distance. As the peptide segments consecutively pair up during folding, the effective intra-chain distance decreases. In case matching peptide segments are connected by multiple paths, the number of segments separating the pairing termini is calculated for all paths, and the shortest path is taken as the intra-chain distance.

The explanation has been added to the main text (page 8). Additionally, a detailed Scheme with an illustrated example of intra-chain distance calculation has been included in the Supplementary Discussion section of Supplementary Information.

(2) Figure 1 is too complicated to follow. Please divide it into some figures and a table in an easy-to-understand manner. In addition, it is difficult for me to understand the graph in the lower panel of Figure 1a. What are the definitions of “contact order” and “effective distance” in this manuscript? How did you calculate them? Please give clear and detailed explanations.

Figure 1 has now been separated into two figures (Figure 1 and Figure 2). Figure 1 now includes a depiction of the CCPO design strategy, contains information on CC modules that are used as building blocks in CCPO cages as well as a description of the nomenclature employed for naming the designed tetrahedron cage variants as suggested by Reviewer #2. Figure 2, on the other hand, now only depicts the modular folding mechanism of TET12SN.

Terms “Contact order” and “effective distance” have been used interchangeably with “intra-chain distance”. To avoid confusion, the former two terms have been replaced with the intra-chain distance in the graph in Fig. 2c and throughout the text.

(3) In Figure 2, the authors should show the results of TET12SN as a control.

Results for TET12SN have been added to the figure (now Figure 3).

(4) The authors should add some movies of typical results of MD simulations in supplementary information.

An example of folding pathways observed during folding simulations of TET12SN are shown in Supplementary Figure 2. Additionally, movies of representative MD simulations for all investigated CCPO tetrahedron variants shown in Supplementary Figure 1 have been included as Supplementary Material.

(5) Please add page numbers in the manuscript.

The numbers have been added. We apologize for the oversight.

(6) In the lower panel of Supplementary Figure 1, how did you figure out which pair was observed to be first to fold?

The G \ddot{o} force field allows protein folding mechanisms to be examined at reduced computational cost by including an attractive term in the energy function to describe non-bonded interactions between atom pairs located in close proximity in the native structure. Atom pairs forming native contacts were determined based on the SAXS-validated CCPO model structures (Ljubetič, A. *et al. Nat. Biotechnol.* **35**, 1094–1101 (2017)) using the so-called Shadow algorithm (Noel, J. K., Whitford, P. C. & Onuchic, J. N. *J. Phys. Chem. B* **116**, 8692–8702 (2012)). During folding simulations, a CC module was considered formed once 50 % or more of its native contacts were reconstituted.

Additional description of the G \ddot{o} force field and the results of MD simulations has been included in the main text (page 8) and in the caption to Supplementary Figure 1.

We thank the reviewer for his comments that helped us to improve the clarity of the manuscript.

Reviewer #2 (Remarks to the Author):

The authors provide an interesting set of experimental and computational data, investigating the folding pathway for a designed coiled-coil tetrahedral protein. They develop a model for the folding pathway base only on the inter-chain distance between complementary coils, and use this to design simplified sequences, which are still able to fold into the desired structures, as determined by SAXS. I think the greatest strength of the paper is the use of the developed model in the successful design of the simplified sequences. However, the work could be greatly strengthened with the addition of some key experiments / controls which would allow the authors to more fully substantiate their folding model. I think this is interesting work and may be publishable in Nature Communications after the authors address the following questions / comments:

1) It would be really useful to see some additional experimental investigation of the proposed folding pathway. This would help to convince readers of the sequential nature of the folding process. Additionally some more detailed discussion of the folding process itself, with respect to the initial formation of helices, and their docking, would be helpful: e.g. it was not clear to me if the authors are proposing a classical framework model in which all the alpha helices form first, and then the closest complementary helices dock, or if the helices form cooperatively upon docking, but do this in sequence. This distinction between secondary structure formation and docking of local helices is important and needs further comment in the main text.

I list here some experiments which would provide useful information, though I understand in the

current Covid climate that some / none of these may be possible to perform at this time, in which case some discussion of the current limitations of the work should be included in the main text.

a) An equilibrium chemical denaturation curve followed by CD, FRET or intrinsic fluorescence (if possible) and preferably all three, or reference to these if it is already in the literature. This may allow the authors to detect equilibrium (un)folding intermediates not populated under thermal denaturation.

The suggested equilibrium chemical denaturation measurements were performed and their outcome is shown in Supplementary Figure 5. The unfolding of TET12SN was monitored by CD and FRET. Additionally, we monitored intrinsic fluorescence of TET12SN as a function of Gdn-HCl concentration, however since Trp and Tyr residues are solvent exposed in TET12SN, only a weak change in intrinsic fluorescence was observed. The data were therefore not included in the manuscript.

The chemical denaturation profile obtained via CD suggested TET12SN undergoes a cooperative two-state unfolding with a denaturation midpoint at 2.5 M Gdn-HCl (Supplementary Fig. 5a). On the other hand, denaturation experiments with fluorescently labelled TET12SN variants revealed that the decrease in helical content is somewhat preceded by a loss of tertiary structure (Supplementary Fig. 5b). For most cage edges, the denaturation midpoint was observed at 2 M Gdn-HCl. However, the cage edge represented by the BCRSN module exhibited a transition midpoint at 1.5 M Gdn-HCl, while the unfolding transition of the APHSN module was less cooperative with a midpoint at approximately 3 M Gdn-HCl. Taken together, the results suggest unfolding of TET12SN is a multi-step process.

Regarding the relationship between CC assembly and helix formation, equilibrium denaturation measurements suggest that while helical content partially increases before CC assembly, the main increase occurs concomitantly with CC formation.

These results are described and discussed in the main text (pages 9 and 21).

Also, it is crucial to show that the protein is fully folded in 2 M Gdn-HCl, the conditions for much of the kinetic refolding studies.

Equilibrium chemical denaturation measurements revealed 2 M Gdn-HCl corresponds to the denaturant concentration where the unfolding transition of TET12SN is observed. To avoid overlooking any less stable intermediate, stopped-flow experiments shown in Figure 2a were repeated at a final Gdn-HCl concentration of 1 M (Supplementary Fig. 7). The temporal order of CC module formation was predominantly unchanged, however the folding constant of the APHSN module could not be reliably determined (Supplementary Fig. 8). The observed folding constant for the cage edge represented by the APHSN module was lower in 1 M Gdn-HCl than in 2 M Gdn-HCl. This so-called roll-over effect was investigated in more detail (Supplementary Fig. 8). For several reasons explained in the Supplementary Discussion section of Supplementary Information we posit the roll-over effect is only apparent and a consequence of unreliable folding constant determination due to the rapid folding of the APHSN module.

b) Ideally the thermal denaturation should be repeated with both heating and cooling cycles, to show the complete thermodynamic reversibility of this process.

Thermal denaturation scans were repeated and show the unfolding is reversible (Supplementary Figure 12). A comment has been added to page 15 of the main text.

c) Stopped-flow CD to monitor the un/refolding process might also yield insights into the sequential model, as refolding when followed by CD (rather than by FRET) may be a multi-state process, especially as two distinct transitions are seen in the thermal denaturation of some designed proteins (e.g. Fig 3).

Refolding of TET12SN was monitored by stopped-flow CD previously (Ljubetič, A. *et al. Nat. Biotechnol.* **35**, 1094–1101 (2017)) and suggested that this protein folds in a two-state process. Global refolding most likely appears to follow an apparent two-state process due to insufficient resolution between folding transitions of individual CC modules. A reference to the previous work was included on page 9 of the main text.

To explain the two transitions observed in thermal denaturation profiles of some tetrahedral cages, we monitored FRET during thermal denaturation of TET12SN (Supplementary Fig. 6). While the CD thermal denaturation profile of TET12SN pointed to a two-state unfolding, FRET experiments demonstrated that individual cage edges have different thermal stabilities. Therefore, we posit that the observed changes in the number of unfolding transitions are due to differences in the constituent CC module composition. Including multiple copies of the same CC module, increases its relative contribution to the unfolding profile leading to individual edge unfolding transitions becoming more resolved. For example, in case of TET12SN(222CC) two transition midpoints are observed at 44 °C and 56 °C (Figure 4g) with the first corresponding well to the melting temperature of the main unfolding transition of the APHSN module observed in the context of the tetrahedral cage (Supplementary Fig. 6).

Thermal denaturation of TET12SN monitored by FRET is shown in Supplementary Fig. 6 and discussed in the main text on pages 9-10. The observed differences in thermal denaturation profiles of different cage variants are commented on in the main text on pages 15 and 16.

Minor points from the text:

2) Consider adding an introductory figure with the protein structure and nomenclature as a main text figure 1 – rather than the first figure call out being to a supplementary figure.

The suggested figure was included in the manuscript as Figure 1. The figure depicts interactions in parallel and antiparallel CC dimer (panel a), CCPO design strategy (panel b) and contains information on CC modules that are used as building blocks in CCPO cages (panel c, d). Explanation of the nomenclature was provided in panel e.

3) The intra-distance measurement needs to be more clearly defined. How is this calculation done?

Due to the modularity of CCPO cages, we introduced the intra-chain distance as a simple metric for describing the changing proximity of pairing peptide segments during folding of CCPO cages. We defined intra-chain distance as the minimal number of peptide modules separating the termini of complementary peptide segments. For each CC module, two distances are calculated. For antiparallel CCs the distance is calculated between the opposing termini (N-C), while for parallel CCs, matching termini are used for the distance calculation (N-N and C-C). In both cases, the minimum of the two values is taken as the intra-chain distance. As the peptide segments pair up in consecutive folding steps, the effective intra-chain distances decrease. In case matching peptide segments are connected by multiple paths, the number of segments separating pairing termini is calculated for all paths, and the shortest path is taken as the intra-chain distance.

The explanation has been added to the main text (page 8). Additionally, a detailed Scheme with an illustrated example of intra-chain distance calculation has been included in the Supplementary Discussion section of Supplementary Information.

4) Fig S5a is somewhat confusing. It also seems to imply formation of the helices occurs concurrently with their docking, but this is not discussed (see 1 above).

As described above, the comparison of equilibrium fluorescence and CD spectra at different Gdn-HCl concentrations (Supplementary Figure 5) indicated that while there is a slight increase in the helical content before CC assembly the majority of α -helical structure is formed upon tertiary structure formation.

A comment on the secondary structure formation has been included in the main text (page 9 and 21).

5) Fig 2a,b: Please define the black lines linking various helices in the schematics under the structures.

The black lines were used to highlight on-target peptide pairings in case of reoccurring CC building blocks. The explanation was added to the caption.

6) Fig 2g,h: It is difficult to see the refolded (red) trace behind the blue. Consider offsetting?

The SEC-MALS traces have been separated into the upper (before refolding) and lower (after refolding) panel in Figures 3 and 4.

7) The appearance of two transitions in the thermal denaturations in Figure 3 is interesting and worthy of discussion: what is the interpretation of this, compared for the single transitions present in Fig 2, and why does this difference occur?

Explained above under the point 1)c).

8) The discussion starts by implying that historically protein folding was thought to be a two-state, cooperative process. This was in fact a new discovery in the 1990's by Jackson, Fersht and others. Rather the historical perspective assumed a multistate process, with hierarchical (framework) models, and hydrophobic collapse as the predominant theories. Nucleation-condensation, two-state mechanisms, and folding funnels were all late to the game! Please revise accordingly.

The reviewer is correct that protein folding was initially thought to proceed through multiple intermediate structures. We apologize for the confusing wording and have clarified the paragraph (page 20).

We would like to thank the reviewer for his comments and suggestions aimed at strengthening our folding model.

Timothy D Craggs
University of Sheffield

Reviewer #3 (Remarks to the Author):

The ability to design synthetic peptide sequences whose interactions and therefore final quaternary structures are predictable is a dream of protein nanoscience. The concept that this could be done using relatively simple interaction rules, somewhat akin to those applied in the DNA origami technique is very interesting and is something the Jerala group have pioneered using coiled-coil interactions. In this work the authors have attempted to understand the folding pathway of such protein origami designs and shown that it proceeds in a stepwise manner with interacting domains that are closest to each other pairing first. This is an original insight and with this knowledge they are also able to show that more than one pair of identical interacting domains can be employed in the same structure so long as the distance rules are adhered to. These results provide useful information for protein designers who may be able to use these findings in their own designs. Furthermore it may

provide some insights into how natural multi-domain proteins fold.

Overall the work is of high quality and clearly presented. However we have identified a number of points both major and minor that require correction or improvement.

Major points

-Page 4, The Gō model is mentioned. For non-experts (like me) there is insufficient explanation of what this model is or how it works. A cursory explanation in the text would be very useful, particularly given the general readership of Nat. Comm.

The Gō force field allows protein folding mechanisms to be examined at reduced computational cost by including an attractive term in the energy function to describe non-bonded interactions between atom pairs located in close proximity in the native structure. Atom pairs that form native contacts were determined based on the SAXS-validated CCPO model structures (Ljubetič, A. *et al. Nat. Biotechnol.* **35**, 1094–1101 (2017)) using the so-called Shadow algorithm (Noel, J. K., Whitford, P. C. & Onuchic, J. N. *J. Phys. Chem. B* **116**, 8692–8702 (2012)).

A brief description of the advantages of the Gō force field has been included in the main text (page 8).

-When introducing the concept of interchain distance it should be made clear that whether the two coiled-coil modules are parallel or antiparallel makes a difference to this calculation. At first this is not obvious when looking at figures

Indeed, the intra-chain distance is calculated differently for parallel and antiparallel CC pairs. We defined intra-chain distance as the minimal number of peptide segments between termini of pairing CC segments. In case of parallel CCs the distance is calculated between matching termini (C-C or N-N), while for antiparallel CCs the distance is calculated between opposing segment ends.

The explanation has been added to the main text (page 8). Additionally, a detailed Scheme with an illustrated example of intra-chain distance calculation has been included in the Supplementary Discussion section of Supplementary Information.

-Page 6 "Additionally, we observed that coiled-coil forming peptides positioned at either terminal end exhibited a slightly increased folding speed". This phrase is unclear and requires further explanation. Do the authors mean that if we compare two folding pairs with the same distance between them but one is closer to the terminus than the other, then the closer one folds quicker? This seems to be suggested from close inspection of Supp. Fig.1 but it should be explained more clearly and better indicated in the figures exactly which pairs demonstrate this point.

The reviewer's assumptions are correct. Coiled-coil modules, that had one peptide partner positioned at the N- or C-terminal end exhibited a slightly higher folding speed than would be expected based on their intra-chain distance. For example in Supplementary Figure 1, while APH4 and P5:P6 modules shared the same intra-chain distance in TET12(2.3)SN-f5b, pair P5SN:P6SN, with peptide P5SN positioned at the N-terminal end, was four-times more likely to form first.

The effect of terminal positioning on the folding kinetics observed in MD simulations was elaborated upon in the main text (page 8) and Supplementary Figure 1.

-Figure 1 Legend "In each step, the peptide pair with the shortest inter-pair distance forms". But in the figure there are 3 possible pairs that have a contact order of 3 (APHSN, GCNSN and P5SN:P6SN).

Why is one shown happening first in preference to the others. Are there in fact three pathways? If so this should be indicated and explained.

Figure 2 (before Figure 1) depicts the proposed modular folding mechanism of TET12SN. The pathway shown in panel b has been suggested based on the order in which CC modules assemble in TET12SN, which was determined with stopped-flow FRET experiments (panel a). We observed that at each step, the forming CC edge had or was among those with the shortest intra-chain distance between its peptide building blocks, indicating that the distance between CC segments plays a dominant role in determining the folding pathway. However, at certain steps, as the reviewer has correctly observed, several pairs have the same intra-chain distance. For example, while APHSN is the first to fold in TET12SN (Figure 2), three CC modules share the same intra-chain distance. We posit the latter could be due to the higher thermodynamic stability of APHSN ($T_m > 80$ °C), however why certain CC pairs form first when the intra-chain distance is kept constant remains to be examined. While multiple pathways were observed in our MD simulations, this information cannot be obtained from the ensemble stopped-flow experiments. However, in the folding model used for designing CCPO cages with repeating CC modules, the folding pathway is indeed split if multiple pairs share the same intra-chain distance and all paths are considered when calculating the probability of correct folding.

The caption to Figure 2 has been expanded to note that several peptide pairs can share the same intra-chain distance. The discussion has been included in the main text (page 21).

-Figure 1 d) Differences in melting temperatures are shown. Could the authors include an explanation for the observed differences?

The observed differences in melting temperatures of CC building blocks are due to the different number of polar residues in the hydrophobic core and different number of salt bridges between residues at *e*, *g* heptad positions. Homodimers APHSN, BCRSN and GCNSN all contain a single polar residue in their hydrophobic interface. Additionally, in APHSN all *e:g* pairs can participate in the formation of salt bridges, resulting in its high melting temperature (> 80 °C). Both BCRSN and GCNSN exhibited lower thermal stability ($T_m = 50$ °C and 33 °C, respectively), due to a lower number of attractive *e:g* interactions and, in case of GCNSN, reduced peptide length. On the other hand, peptide pairs P3SN:P4SN, P5SN:P6SN and P7SN:P8SN ($T_m = 33$ °C, 28 °C and < 20 °C, respectively) all contain two energetically unfavourable Asn-Asn pairings in the hydrophobic interface, while all *e:g* polar interactions are complementary. Differences in their thermal stabilities can be explained by considering the residue pattern at position *a* and electrostatic pattern at *e* and *g* positions (in each heptad positions *e* and *g* are occupied by the same residue). It has been previously shown that positioning Asn residues at the *a* site in the third and fourth heptad leads to CC assemblies with higher thermodynamic stabilities (IINN $>$ NINI \approx ININ). Additionally, CCs with equally charged neighbouring heptads demonstrate higher melting temperatures (EEKK $>$ EKKE $>$ EKEK). These two rankings match perfectly with the observed order of T_{ms} (P3SN:P4SN $>$ P5SN:P6SN $>$ P7SN:P8SN).

A brief explanation for the differences in T_m has been included in the main text (page 9) with more details given in the Supplementary Discussion section of Supplementary Information.

-Page 13: It is mentioned that TET12SN(3CC)-neg exhibited substantial aggregation in refolding in line with the suboptimal folding as predicted by the model. Can the authors explain then why, if this arrangement had the lowest PF it is not surprising that it was normally expressed, purified and folded before unfolding experiments took place? It actually looks reasonable on SDS PAGE in Supp. Fig. 7 and SEC (Supp. Fig. 10)

While all proteins could indeed be obtained in monomeric form, the isolation yield for both TET12SN(3CC) and TET12SN(3CC)-neg was markedly reduced in comparison to other designed

variants. Moreover, the fraction of oligomeric species in the SEC profile of TET12SN(3CC)-neg was larger than for TET12SN(3CC) (Supplementary Figure 13).

SEC profiles of TET12SN(222CC), TET12SN(3CC) and TET12SN(3CC)-neg after Ni-NTA chromatography have been included in Supplementary Figure 13. A comment has been included in the main text (page 16).

-Page 20: "Relaxation of the requirement for orthogonal sets of parallel and antiparallel CC dimers importantly increases the complexity of CCPO cages that can be achieved with an available orthogonal coiled-coil set." Perhaps the authors could give an example of what structures one is presumably now able to design given the available CC domains and the 50 % decrease of unique domains needed?

Discussion on which polyhedral shapes could be potentially achieved was added to the main text (page 23).

Currently, our validated CC building block set consists of 10 CC modules, which limits the designability to polyhedral objects composed of maximum 10 edges. Using each module twice in CCPO construction will allow us to attempt cage designs resembling polyhedral shapes with up to 20 edges, such as octahedron, pentagonal bipyramid, snub disphenoid or elongated triangular bipyramid.

-All of the work presented has been done with a tetrahedron shape. Can the authors include data to show whether or not this applies to other shapes?

In addition to tetrahedral cages, the proposed mathematical folding model was applied to the design of a cage in the shape of a trigonal bipyramid. The cage contained 9 edges and was composed of 6 unique CC dimers, 3 of which were used twice in the design. The work is part of a separate publication currently undergoing review. To facilitate the review process, a copy of the manuscript was provided to the editor.

A reference to the work was added to the main text (page 23).

-CC dimers are insufficiently introduced and explained. Notably peptide synthesis methods are completely lacking

A passage introducing CC dimers and explaining their interaction patterns has been added to the main text (page 3). Additionally, a helical wheel representation of parallel and antiparallel CC dimers has been included in Figure 1.

Synthetic peptides were purchased from Proteogenix (France). This information has been added to the Methods section of the main text (page 30).

-Supplementary Figure 5: Here the authors introduced additional dotted CC to the scheme which appears to be in the case of blue either P3SN or P4SN and in the case of yellow P5SN and P6SN. Presumably this is meant to show the identical CC copies but this is not explained anywhere. Please explain in the legend

The reviewers' assumptions are correct. The dotted CCs in Supplementary Figure 9 were used to denote different CC module pair copies, in order to easily differentiate between productive (i.e. those leading to a correctly folded tetrahedral cage) and unproductive peptide pairings.

The explanation has been added to the figure caption.

Minor points

-Figure 1 a, Colour of the brown arrow and bar does not match

The colours were changed to match.

-Figure 1a, figure legend. Change "inter-pair" to "intra-chain" as in the rest of the text

“Inter-pair” was changed to “intra-chain”.

-On page 4, the authors refer to "contact order". Are contact order, intra chain distance and inter peptide distance all describing the same? The authors should define one and use throughout the text & figures. This would make the text easier to understand

Indeed the terms were used interchangeably to avoid repetition. In order to improve clarity, “intra-chain distance” is now used throughout the manuscript. A definition of the term was provided in the main text (page 8) with two examples of intra-chain distance calculation provided in the Scheme in the Supplementary Discussion section of Supplementary Information.

-Page 13, line 2. Change "asses" to "assess"

The typo was removed.

- When folding times are mentioned it would be interesting to know the absolute folding times.

The values of refolding constants observed for TET12SN cages in 2 M Gdn-HCl have been added to the main text (page 10).

- Supp. Figure 1: What is the total number of runs of this simulation/ total number of occurrences? Please add this to the figure legend, so the reader does not need to look for it in the methods part.

For each CCPO tetrahedron variants 100 runs were performed, each consisting of 30 million steps. The information has been added to the caption to Supplementary Figure 1.

-Supp. Table 1. Please define the meaning of the lower and upper case letters

In the segments column of Supplementary Table 2 (before Supp. Table 1), same letters denote matching peptide pairs. Same-cased letters signify parallel CC pairs, while different-case letters mark antiparallel CCs as used in the previous publication (Ljubetič, A. *et al. Nat. Biotechnol.* **35**, 1094–1101 (2017)).

The above definition has been added to the caption.

We thank the reviewer for his kind comments and suggestions that contributed towards the quality and readability of the manuscript.

REVIEWERS' COMMENTS

Reviewer #1 (Remarks to the Author):

In this article, the authors demonstrate that the designed coiled-coil protein origami (CCPO) folds in a stepwise sequential pathway. MD simulations and stopped-flow FRET measurements revealed that CCPO folding is dominated by the effective intra-chain distance between CC modules in the primary sequence and subsequent folding intermediates. The number of orthogonal modules required for constructing a CCPO tetrahedron could be reduced from six to as little as three different CC modules. The manuscript includes interesting results and an original concept about folding of CCPO.

The revised manuscript with detailed explanations has been considerably improved. Now I can recommend this manuscript for publication in Nature Communications after the following minor revisions.

(1) P. 2, line 7 in Abstract: I suggest a change from "effective distance" to "effective intra-chain distance".

(2) The authors should add legends for Supplementary Movies in Supplementary Information and refer to the Supplementary Movies in the main text.

(3) I think one of key points of the definition of intra-chain distance is that "assembled CCs are counted as 1 segment", and the authors should also add that in the main text.

Reviewer #2 (Remarks to the Author):

The authors have made considerable changes to the manuscript, including results from all the requested additional experiments, and rewording of key sections.

I am happy for this now to be published.

Timothy D. Craggs
University of Sheffield

Reviewer #3 (Remarks to the Author):

The authors have made considerable efforts to address all points raised. I recommend publication.

Please note two additional minor points:

Second figure of Supp. Figures does not have a figure number

Please check newly added text for English

RESPONSE TO REVIEWERS' COMMENTS

Reviewer #1 (Remarks to the Author):

In this article, the authors demonstrate that the designed coiled-coil protein origami (CCPO) folds in a stepwise sequential pathway. MD simulations and stopped-flow FRET measurements revealed that CCPO folding is dominated by the effective intra-chain distance between CC modules in the primary sequence and subsequent folding intermediates. The number of orthogonal modules required for constructing a CCPO tetrahedron could be reduced from six to as little as three different CC modules. The manuscript includes interesting results and an original concept about folding of CCPO. The revised manuscript with detailed explanations has been considerably improved. Now I can recommend this manuscript for publication in Nature Communications after the following minor revisions.

(1) P. 2, line 7 in Abstract: I suggest a change from “effective distance” to “effective intra-chain distance”.

The suggested change has been inserted.

(2) The authors should add legends for Supplementary Movies in Supplementary Information and refer to the Supplementary Movies in the main text.

We have added a reference to Supplementary Movies in the main text (page 8). Legends for Supplementary Movies have been provided in the author checklist and not in the Supplementary Information file in accordance with the editor’s instructions.

(3) I think one of key points of the definition of intra-chain distance is that “assembled CCs are counted as 1 segment”, and the authors should also add that in the main text.

The information has been added to the main text (page 8).

Reviewer #2 (Remarks to the Author):

The authors have made considerable changes to the manuscript, including results from all the requested additional experiments, and rewording of key sections.

I am happy for this now to be published.

Timothy D. Craggs
University of Sheffield

Reviewer #3 (Remarks to the Author):

The authors have made considerable efforts to address all points raised. I recommend publication. Please note two additional minor points:
Second figure of Supp. Figures does not have a figure number

The figure has been moved and appropriately labelled. It now appears in the Supplementary Information file as Supplementary Figure 10.

Please check newly added text for English.

The added sections have been checked again.

We thank all of the reviewers for the positive feedback and helpful comments.